# An immunohistochemical prostate cell identification key indicates that aging shifts procollagen 1A1 production from myofibroblasts to fibroblasts in dogs prone to prostate-related urinary dysfunction

**Hannah Ruetten** [1,2◦], **Clara Cole** [1,2◦], **Marlyse Wehber** [1,2◦], **Kyle A. Wegner** [2,3], **Nicholas M. Girardi** [2,3], **Nelson T. Peterson** [2,3], **Brandon R. Scharpf** [2,3], **Michael F. Romero** [4], **Michael W. Wood** [5], **Sara A. Colopy** [6], **Dale E. Bjorling** [6], **Chad M. Vezina** [1,2,3]*

1 Department of Comparative Biomedical Sciences, School of Veterinary Medicine, University of Wisconsin-Madison, Madison, Wisconsin, United States of America, 2 George M. O'Brien Benign Urology Center, University of Wisconsin- Madison, Madison, Wisconsin, United States of America, 3 Molecular and Environmental Toxicology Center, School of Medicine and Public Health, University of Wisconsin- Madison, Madison, Wisconsin, United States of America, 4 Physiology and Biomedical Engineering and Nephrology and Hypertension, George M. O'Brien Urology Research Center, Mayo Clinic College of Medicine and Science, Rochester, Minnesota, United States of America, 5 Department of Medical Sciences, School of Veterinary Medicine, University of Wisconsin- Madison, Madison, Wisconsin, United States of America, 6 Department of Surgical Sciences, School of Veterinary Medicine, University of Wisconsin- Madison, Madison, Wisconsin, United States of America

◦ These authors contributed equally to this work.
* cmvezina@wisc.edu

**Data Availability Statement:** The data underlying the results presented in the study are freely

## Abstract

### Background

The identity and spatial distribution of prostatic cell types has been determined in humans but not in dogs, even though aging- and prostate-related voiding disorders are common in both species and mechanistic factors, such as prostatic collagen accumulation, appear to be shared between species. In this publication we characterize the regional distribution of prostatic cell types in the young intact dog to enable comparisons with human and mice and we examine how the cellular source of procollagen 1A1 changes with age in intact male dogs.

### Methods

A multichotomous decision tree involving sequential immunohistochemical stains was validated for use in dog and used to identify specific prostatic cell types and determine their distribution in the capsule, peripheral, periurethral and urethral regions of the young intact canine prostate. Prostatic cells identified using this technique include perivascular smooth muscle cells, pericytes, endothelial cells, luminal, intermediate, and basal epithelial cells, neuroendocrine cells, myofibroblasts, fibroblasts, fibrocytes, and other hematolymphoid cells. To enhance rigor and transparency, all high resolution images (representative images

accessible through the GUDMAP database at https://doi.org/10.25548/16-WMM4.

**Funding:** Funded by National Institutes of Health grants: U54DK104310 (HR, CC, MW, KW, NG, NP, BS, DB, CV), Summer Program in Undergraduate Urologic Research (U54DK104310S1; NG), U54DK100227 (HR, CC, MW, KW, NG, NP, BS, DB, CV), K12DK100022 (MW), T32ES007015 (KW), F31ES028594 (KW), TL1TR002375 (HR), F30DK122686 (HR), and the Mayo Foundation (MR). This content is solely the responsibility of the authors and does not necessarily represent the official views of the National Institutes of Health. The funders had no role in study design, data collection and analysis, decision to publish, or preparation of the manuscript.

**Competing interests:** The authors have declared that no competing interests exist.

shown in the figures and biological replicates) are available through the GUDMAP database at https://doi.org/10.25548/16-WMM4.

## Results

The prostatic peripheral region harbors the largest proportion of epithelial cells. Aging does not change the density of hematolymphoid cells, fibroblasts, and myofibroblasts in the peripheral region or in the fibromuscular capsule, regions where we previously observed aging- and androgen-mediated increases in prostatic collagen abundance Instead, we observed aging-related changes the procollagen 1A1 positive prostatic cell identity from a myofibroblast to a fibroblast.

## Conclusions

Hematolymphoid cells and myofibroblasts are often identified as sources of collagen in tissues prone to aging-related fibrosis. We show that these are not the likely sources of pathological collagen synthesis in older intact male dogs. Instead, we identify an aging-related shift in the prostatic cell type producing procollagen 1A1 that will help direct development of cell type and prostate appropriate therapeutics for collagen accumulation.

## 1. Introduction

Men and dogs develop prostatic disorders spontaneously and in an aging- and androgen-dependent fashion. [1–11] Human and dog prostates organize into distinct anatomical zones with an externally bound fibromuscular capsule (anatomical features not shared by the mouse prostate). [12] Careful histological and molecular analyses of human prostate have revolutionized our understanding of the cell types of the prostate, revealing a gland that contains at least five epithelial cell types and three fibroblast-like stromal cell types that are organized into four zones. [13, 14] Canine prostate cell types and their spatial distribution across the gland have not been extensively examined using modern molecular and multiplex immunohistochemical methods. This void of knowledge hinders the translational potential of canine prostate findings to human medicine in a model species that bolsters great benefit to the prostate research field.

Our study utilizes young intact male dog prostate specimens to develop a regional atlas of prostatic epithelial and stromal cell types to enable future cross-species comparisons among dog, human and mouse models. We found that the dog prostate is anatomically divided into the capsule, peripheral, periurethral and urethral regions and that the distribution of prostatic epithelial and stromal cell types is not homogenous but distinct to each region. Next, we applied our new knowledge of canine prostate cell types and their spatial distribution and determined how the cellular source of procollagen 1A1 changes with age in intact male dogs.

Prostatism is a syndrome in aging males characterized by lower urinary tract symptoms (LUTS) that are responsive to medical and surgical therapies targeting the prostate.[1] It was previously hypothesized that an enlarged prostate gland of non-neoplastic origin (benign prostatic hyperplasia, benign prostatic hypertrophy, and/or benign prostatic enlargement) was solely responsible for the syndrome.[2-5] However, men with enlarged prostates can be asymptomatic and prostate size is weakly correlated with symptom severity.[6,7] Recent studies show stronger correlations among prostatic inflammation (prostatitis), collagen accumulation (fibrosis), and LUTS severity. [8–16]

We previously used whole transverse sections of canine prostate to map aging- and castration-mediated changes in prostatic collagen architecture. [12] We found that aging increases collagen fiber thickness and density in the capsule and peripheral regions of intact dogs. [12] Numerous studies across multiple organ systems have identified myofibroblasts as fibrosis inducing cell types. [15–20] The second goal of this study is to test if myofibroblast populations expand and produce pro-collagen type 1A1 in the regions of the canine prostate prone to aging related collagen accumulation. We found that aging does not significantly alter the distribution of hematolymphoid cells (defined as cells immunopositive for protein tyrosine phosphatase receptor type C, also known as CD45), myofibroblasts (defined as cells immunopositive for vimentin (VIM), actin alpha 2 (ACTA2+) and S100 calcium binding protein A4 (S100A4, also known as fibroblast specific protein 1). Aging does not significantly change abundance of myofibroblasts, hematolymphoid or pro-collagen 1A1 positive cells. Instead, aging shifts the identity of procollagen 1A1 producing cells from predominantly ACTA2+, S100A4+ myofibroblasts to predominately ACTA2-, S100A4+ fibroblasts. Our results will help direct development of appropriate therapeutics for collagen accumulation in aged intact males.

## 2. Materials and methods

### 2.1. Prostatic tissue collection, preparation, and processing

Canine prostatic tissue sources were previously described. [12] Prostates were collected postmortem from dogs euthanized for reasons other than prostatic disease. Prostates were from the Dane County Humane Society (Madison, WI), the University of Wisconsin-Madison Veterinary Teaching Hospital Necropsy Service (Madison, WI), Covance (Madison, WI), and the Mayo Clinic (Rochester, MN). Dogs ranged in age from 0.83 years to 15 years with a mean of 5.95 years and median of 5 years. They ranged in weight from 4.82 to 47.10 kg with a mean of 27.02 kg and median of 28.5 kg. Weight information was not recorded for six dogs at time of euthanasia. Prostates were obtained from multiple breeds to ensure representation across the canine population: Golden Retriever (4), Beagle (2), Border Collie (2), Pit Bull (2), Boxer (1), Cocker Spaniel (1), Doberman Pinscher (1), French Bulldog (1), Labrador Retriever (1), Miniature Poodle (1), Rat Terrier (1), Rottweiler (1), Shih Tzu (1), Standard Poodle (1), Weimaraner (1), mixed breed (8), and unknown breed (2).

Benign prostatic enlargement occurs in intact dogs but not neutered dogs and is most common in intact dogs >6 years of age but may begin as early as 2.5 years of age. [1] Therefore, to maximize group size and comparisons between young and old, this study included young (≤3 years) and old (≥5 years) intact dogs and excluded middle aged dogs. Intact dogs were defined as those with two palpable testicles present in scrotum at the time of euthanasia. Dogs with unilateral or bilateral cryptorchidism were excluded.

Prostates were fixed and stored in neutral-buffered formalin. Prostates were bisected by a single dorsal-ventral cut at cranial-caudal midline and then cut into slices suitable for embedding. Central slices were oriented in transverse plane, embedded in paraffin, and sectioned by the University of Wisconsin-Madison Veterinary Teaching Hospital Histology Service. Twenty serial sections were taken from each embedded slice. The first and eleventh section were stained with hematoxylin and eosin, and the second and twelfth were stained with picrosirius red. All remaining slides were utilized for immunohistochemistry.

### 2.2. Immunohistochemistry

Immunofluorescence staining was conducted as previously described. [21] Briefly, tissue sections (5 μm) were deparaffinized with xylene and rehydrated in a series of graded ethanols

**Table 1. Primary and secondary antibodies.**

| Name(s) | Symbol | Species | Antibody Registry (RRID) | Supplier | Catalog Number | Dilution | Un-masking | Fig # |
|---|---|---|---|---|---|---|---|---|
| Actin alpha 2 | ACTA2 | Goat | AB 10980764 | Thermo Fisher Sci. | PA5-18292 | 1:100 | Citrate | 2, 3, 4, 5, S2 |
| Platelet and endothelial cell adhesion molecule | PECAM | Mouse | AB 2801330 | Santa Cruz Biotech. | SC-376764 | 1:200 | Tris EDTA | S2 |
| Platelet derived growth factor receptor beta | PDGFRB | Rabbit | AB 2162497 | Cell Signaling Tech. | 3169S | 1:100 | Tris EDTA | S2 |
| Keratin 5 | KRT5 | Chick | AB_2565054 | Biolegend | 905901 | 1:100 | Citrate | 1 |
| Keratin 8/18 | KRT8/18 | Guinea Pig | AB 1284055 | Fitzgerald | 20R-CP004 | 1:500 | Citrate | 1 |
| Chromogranin A | CHGA | Rabbit | AB 301704 | Abcam | ab15160 | 1:200 | Citrate | 1 |
| Vimentin | VIM | Mouse | AB 445527 | Abcam | AB20346 | 1:250 | Citrate | 2, 3, 4 |
| Protein tyrosine phosphatase, receptor type C | PTPRC | Rabbit | AB 442810 | Abcam | AB10558 | 1:500 | Citrate | 2, 4, S3, S4 |
| S100 calcium binding protein A4 | S100A4 | Rabbit | AB_2183775 | Abcam | AB27957 | 1:200 | Citrate | 3, 4, 5 |
| Procollagen type I | ProCOL1A1 | Mouse | N/A | Developmental Studies Hybridoma Bank | SP1.D8-c (concentrated, 238ug/mL) | 1:1000 | Citrate | 5 |
| Anti-mouse Alexa Fluor 647 | - | Donkey | AB_2340846 | Jackson Immuno Res. | 715-545-150 | 1:250 | - | 2, 3, 4, 5, S2 |
| Anti-goat Alexa Fluor RRX | - | Donkey | AB_2340423 | Jackson Immuno Res. | 705-295-147 | 1:250 | - | 2, 3, 4, S2 |
| Anti-rabbit Alexa Fluor 488 | - | Donkey | AB 2313584 | Jackson Immuno Res. | 711-545-152 | 1:250 | - | 1, 2, 3, 4, S2, S3 |
| Anti- chicken Alexa Fluor 594 | - | Donkey | AB_2340377 | Jackson Immuno Res. | 703-585-155 | 1:250 | - | 1 |
| Anti- guinea pig Alexa Fluor 647 | - | Goat | AB_2337446 | Jackson Immuno Res. | 106-605-003 | 1:250 | - | 1 |
| Anti-goat Alexa Fluor 488 | - | Donkey | AB_2340428 | Jackson Immuno Res. | 705-545-003 | 1:250 | - | 5 |
| Anti-rabbit Alexa Fluor 546 | - | Donkey | AB_2534016 | Thermo Fisher Scientific | A10040 | 1:250 | - | 5 |
| Anti-rabbit Alexa Fluor RRX | - | Donkey | AB_2340613 | Jackson Immuno Res. | 711-295-152 | 1:250 | - | S4 |

(100%, 75%, 50%). Microwave decloaking was performed in 10 mM sodium citrate (pH 6.0) and non-specific sites were blocked in TBSTw containing 1% Blocking Reagent (Roche Diagnostics, Indianapolis, IN), 5% normal goat or donkey sera, 1% bovine serum albumin fraction 5. Primary and secondary antibodies are listed in Table 1. Immunofluorescent staining was imaged using a BZ-X710 digital microscope (Keyence, Itasca, IL, USA) fitted with at 20x (Plan-Fluor, NA 0.45) objective. Some tissue sections were imaged using an Eclipse E600 compound microscope (Nikon Instruments Inc., Melville, NY) fitted with a 20x dry objective (Plan Fluor NA = 0.75; Nikon, Melville, NY) and equipped with NIS elements imaging software (Nikon Instruments Inc.). Fluorescence was detected using DAPI (2-(4-amidinophenyl)-1H -indole-6-carboxamidine), FITC, Texas Red (Chroma Technology Corp, Bellows Fall, VT), and CY5 filter cubes (Nikon, Melville, NY).

## 2.3. Cell counting

Two to three randomly selected images were analyzed per region per dog. The capsule is defined as the outer layer of fibromuscular tissue encapsulating the prostate (S1 Fig). The urethral region contains urethral luminal cells as well as their supporting stroma (S1 Fig). The peripheral and periurethral regions are not delineated by a visible anatomic or histologic

boundaries. We defined them by dividing the glandular region into two concentric rings of equal thickness (S1 Fig). Absolute cell counts (all cells containing a nuclei and meeting staining criteria for the cell type) were performed manually using Image J cell counter. [22] All cells were counted in every image (one image = one 20X field; the size of the 20X field for samples imaged using the Nikon microscope is 435.84 μm X 331.42 μm, the 20X field size for the BZ-X710 microscope is 964.19 μm X 723.14 μm). Antibodies against cytokeratin 8/18 (KRT 8/18), cytokeratin 5 (KRT 5), and chromogranin A (CHGA) were used to identify luminal [21, 23] (KRT 8/18+, KRT 5-, CHGA-), intermediate [21, 23] (KRT 8/18+, KRT 5+, CHGA-), and basal [21, 23] (KRT 8/18-, KRT 5-, CHGA-) epithelial cells. Neuroendocrine cells [21, 24, 25] (KRT 8/18-, KRT 5-, CHGA+) were also identified with this stain combination. Antibodies against smooth muscle actin (ACTA2), protein tyrosine phosphatase receptor type C (PTPRC, also known as CD45), and vimentin (VIM) were used to identify fibrocytes [21, 26–30] (ACTA2+, PTPRC+, VIM+), and other hematolymphoid cells (PTPRC+, not a fibrocyte). Antibodies against ACTA2, VIM, and fibroblast specific protein (S100A4, also known as FSP1) were used to identify myofibroblasts [21, 31–34] (ACTA2+, VIM+, S100A4+), and fibroblasts [21, 27, 35–37] (ACTA2-, VIM+, S100A4+). Results are expressed as cells per $mm^2$. Statistical comparisons were made between or among images collected form the same microscope using the same imaging settings and magnification.

## 2.4. Determining distribution of hematolymphoid cells from the urethra to the capsule

Immunohistochemistry, as described in section 2.2, was performed to label hematolymphoid cells using antibodies against protein tyrosine phosphatase receptor type C (PTPRC) and imaged with the BZ-X710 digital microscope (Keyence, Itasca, IL, USA) fitted with the 20x (PlanFluor, NA 0.45) objective. From transverse sections, we collected 20X image tiles and stitched them to produce a continuous image running from capsule to urethra (S4 Fig). The stitched image was rotated so that the urethra was oriented on the left side. The image was opened in image J, the entire image was selected and the tool 'Plot Profile' applied to determine the average gray value for each column of pixels in the image. Gray values were exported as (X, Y) coordinates and used to create a linear plot of inflammatory cell distribution. Images containing PTPRC staining were overlaid with those containing DAPI and the x-axis of the linear plot of gray values was fit to the x-axis of the new overlay image (S4A–S4D Fig). Areas with more hematolymphoid cells have higher gray values, areas with less hematolymphoid cells have lower gray values.

## 2.5. Statistical analyses

Statistical analyses were performed with Graph Pad Prism 8.0.2 (Graphpad Software, La Jolla, California). The Shapiro-Wilk test was used to test for normality and transformation was applied to normalize data if necessary. The F-test was used to test for homogeneity of variance for pairwise comparisons. Welch's correction was applied when variances were unequal. When variances were equal, Student's t-test was used to test for differences between groups. The Mann Whitney test was applied when data could not be normalized through transformation. One-way analysis of variance (ANOVA) followed by Tukey's post-hoc analysis was used for identifying differences among groups. A p value of less than 0.05 was considered statistically significant.

## 3. Results

### 3.1. Distribution of prostatic cell types in the young intact dog

Historical accounts of canine prostate histology describe an unequal distribution of epithelial and stromal cells, with stromal cells predominating close to the urethra and epithelial cells predominating peripherally. [38] However, modern descriptions have lost this distinction and describe a histologically homogenous gland. [5, 39] We previously published an immunohistochemical key for identifying and objectively quantifying mouse prostatic epithelial, perivascular, and fibromuscular cells. [21] We apply an adapted immunohistochemical key here to test whether dog prostate cell types are homogenously distributed across the gland. The only differences between the prostate cell identification key previously published for mouse [21] and the key applied here for dog are that 1) the mouse key identified two unique populations of 'Other hematolymphoid cells' differentiated by vimentin immunostaining and the dog key combines both populations into a single, 'Other hematolymphoid cell' population, 2) the dog prostate cell identification key excludes an androgen receptor stain because the antibody used previously for mice does not cross react with dog, and 3) Chromogranin A was used in place of synaptophysin to identify neuroendocrine cells. We used epithelial and fibromuscular stains to determine the regional distribution of four epithelial cell types and four stromal cell types in young intact male dog prostate. We also successfully detect endothelial cells, pericytes, and perivascular smooth muscle myocytes as shown in S2 Fig.

**3.1.1. Epithelial cells.** Antibodies against cytokeratin 8/18 (KRT 8/18), cytokeratin 5 (KRT 5), and chromogranin A (CHGA) are used to identify luminal [21, 23] (KRT 8/18+, KRT 5-, CHGA-, Fig 1B), intermediate [21, 23] (KRT 8/18+, KRT 5+, CHGA-, Fig 1C), and basal [21, 23] (KRT 8/18-, KRT 5-, CHGA-, Fig 1D) epithelial cells. Neuroendocrine cells [21, 24, 25] (KRT 8/18-, KRT 5-, CHGA+, Fig 1E) are also identified. Epithelial cells in general and neuroendocrine cells in particular are not detected in the capsule (Fig 1F) but are present in the peripheral region (Fig 1G), the periurethral region (Fig 1H) and the prostatic urethra (Fig 1I). The peripheral region consists of 96% epithelial cells, the periurethral region consists of 67% epithelial cells and the prostatic urethra consists of 33% epithelial cells (Fig 1H). Luminal epithelial cells are absent from the capsule, comprise 100% of epithelial cells in the peripheral region, 96% of epithelial cells in the periurethral region and 67% of epithelial cells in the urethral region (Fig 1I). Basal and intermediate epithelial cells are not detected in the capsule, comprise <1% of epithelial cells in the peripheral region, 4% of epithelial cells in the periurethral region, and 31% of epithelial cells in the urethral region (Fig 1I). Neuroendocrine cells are not detected in the capsule, comprise <1% of epithelial cells in peripheral and perirurethral regions, and comprise 2% of epithelial cells in the urethral region (Fig 1I).

**3.1.2. Stromal cells.** ACTA2, protein tyrosine phosphatase receptor type C (PTPRC), and vimentin (VIM) are used to identify fibrocytes [21, 26–30] (ACTA2+, PTPRC+, VIM+, Fig 2B), and other hematolymphoid cells (PTPRC+, not a fibrocyte, Fig 2C) in canine prostates. Fibrocytes are relatively rare but comparable in frequency across all regions of the prostate (Fig 2D–2G and 2H). The density of 'Other hematolymphoid cells' does not significantly differ across prostatic regions (Fig 2D–2G and 2I).

ACTA2, VIM, and fibroblast specific protein (S100A4) are used to identify myofibroblasts [21, 31–34] (ACTA2+, VIM+, S100A4+, Fig 3B), and fibroblasts [21, 27, 35–37] (ACTA2-, VIM+, S100A4+, Fig 3C) in canine prostates. Fibroblasts are more abundant in the urethra than in the peripheral region and myofibroblasts are less abundant in the peripheral and periurethral than the urethra.

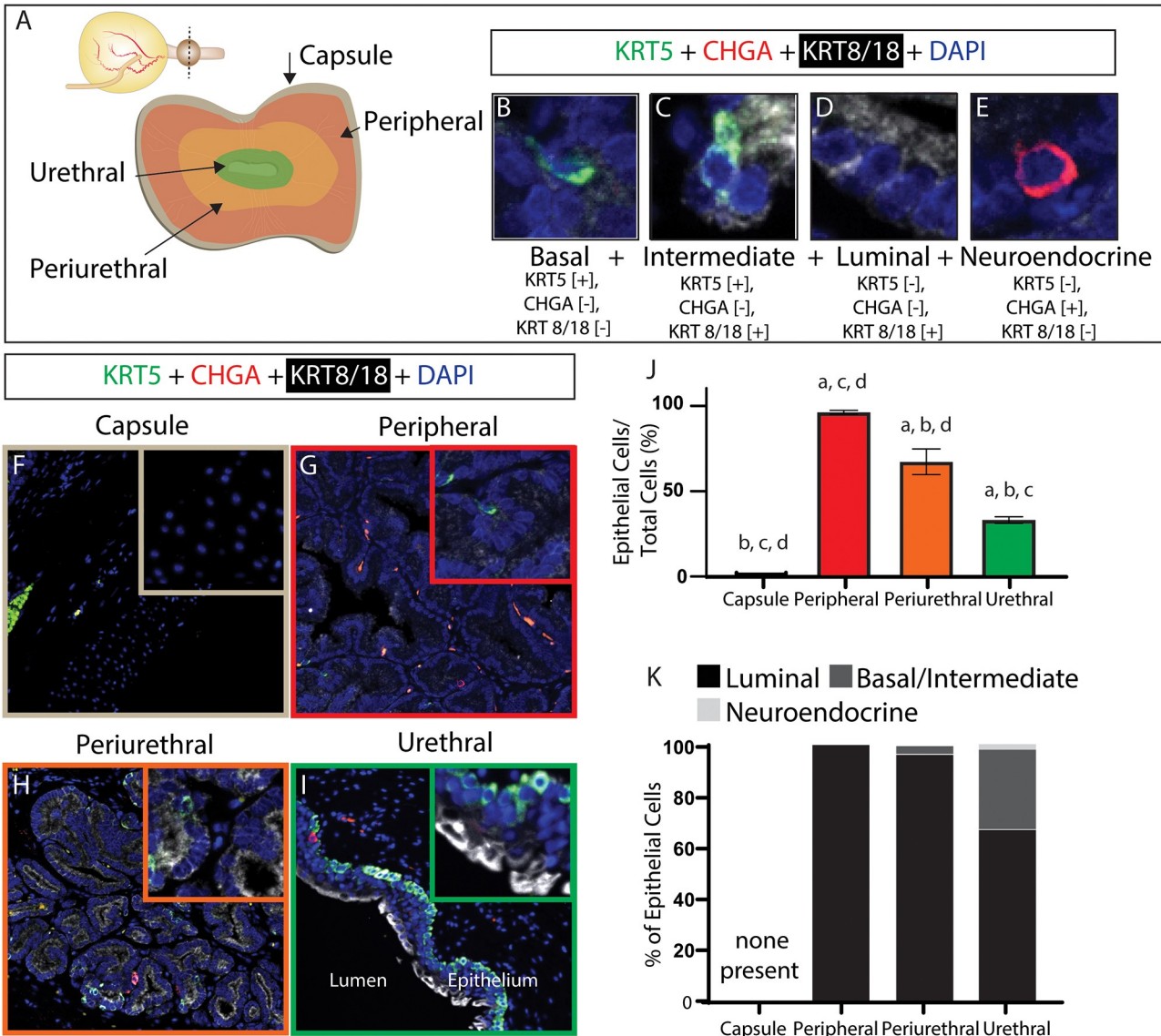

**Fig 1. Young intact canine prostatic epithelial cell composition and distribution.** Multiplex immunostaining with antibodies against Keratin 5 (KRT5), Keratin 8/18 (KRT8/18) and Chromogranin A (CHGA) was performed on complete transverse sections of young (≤3 years old) intact male canine prostate. Nuclei were stained with DAPI. (A) The prostate was divided into capsule, peripheral, periurethral and urethral zones to describe the regional distribution of cells. B-E depict the staining pattern criteria of each quantified cell type with an example cell image. Basal epithelial, intermediate epithelial, luminal epithelial and neuroendocrine cells were identified by multiplex staining. (F-I) Epithelial cells were visualized and densities quantified as a function of (J) all cells and (K) distribution of epithelial cell types within each region. (J) Results are mean ± SE, A one-way ANOVA was used to compare regions, Letters (a-d) indicate that epithelial cell density significantly differs (p<0.05) across regions. Differences between each examined prostatic region and (a) capsule, (b) peripheral, (c) periurethral, (d) urethral regions are identified. (K) Results are mean (%) of total epithelial cells. Images are representative of three young intact male dog prostates. High resolution images contributing to this figure are freely accessible through the GUDMAP database at https://doi.org/10.25548/16-WMM4.

## 3.2. Aging does not change the density of myofibroblasts, fibroblasts, and hematolymphoid cells in the capsule and peripheral region of intact male dogs

We previously identified an aging- and androgen-dependent increase in prostatic collagen density in the prostatic capsule and peripheral regions of intact male canines. [12] We also

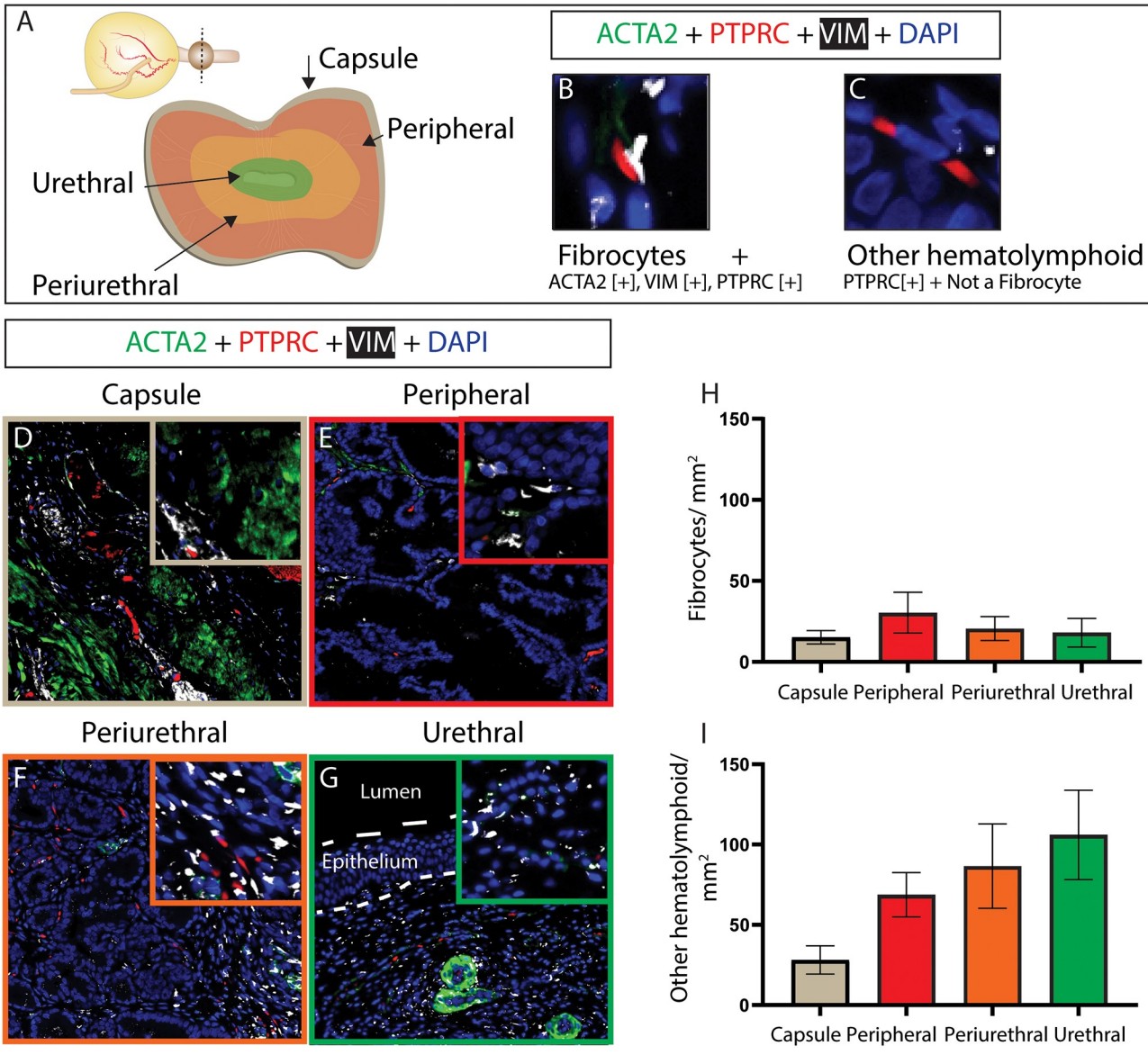

**Fig 2. Young intact canine prostate fibrocyte and other hematolymphoid cell distributions.** Multiplex immunostaining with antibodies against smooth muscle actin (ACTA2), protein tyrosine phosphatase, receptor type C (PTPRC) and vimentin (VIM) was performed on (A) complete transverse sections of young (≤3 years old) intact male canines. Nuclei were stained with DAPI. B-C depict the staining pattern criteria of each quantified cell type with an example cell image. Fibrocytes and other hematolymphoid cells were identified by multiplex staining. (D-I) Fibrocytes and Other hematolymphoid cells were visualized and densities quantified within each region. (H-I) Results are mean ± SE. A one-way ANOVA was used to compare cell densities among regions. There was no significant difference (p<0.05) among groups. Images are representative of 6–8 young intact male dog prostates. High resolution images shown here and others from biological replicates are available through the GUDMAP database at https://doi.org/10.25548/16-WMM4.

compared tissue architecture between young and old intact male dogs across the four prostatic regions (S3 Fig). Here we test the hypothesis that aging increases myofibroblast density in these canine prostate regions.

We stained prostate sections from old intact male dogs and counted myofibroblasts [21, 31–34] and fibroblasts [21, 27, 35–37] in each region. The number myofibroblasts and fibroblast per 20X field in the capsule and peripheral region did not significantly differ between old and young intact males (Fig 4).

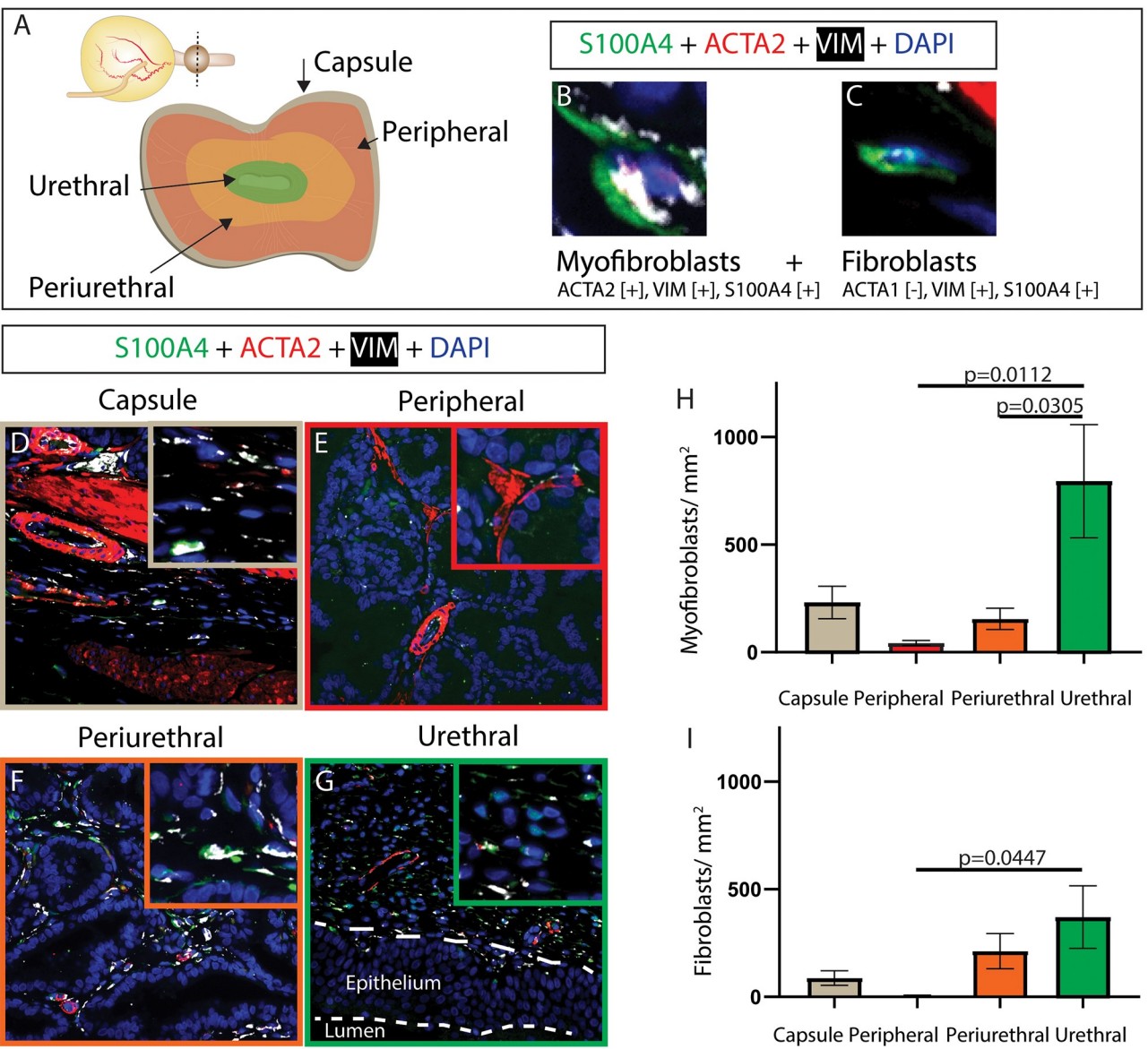

**Fig 3. Young intact canine prostatic myofibroblast and fibroblast distributions.** Multiplex immunostaining with antibodies against smooth muscle actin (ACTA2), fibroblast specific protein (S100A4) and vimentin (VIM) was performed on (A) complete transverse sections of young (≤3 years old) intact male canines. Nuclei were stained with DAPI. B-C depict the staining pattern criteria of each quantified cell type with an example cell image. Fibroblasts and myofibroblasts were identified based by multiplex staining. (D-I) Fibroblasts and myofibroblasts were visualized and densities quantified within each region. (H-I) Results are mean ± SE. A one-way ANOVA was used to compare cell densities among regions. Cell density differences between regions (p <0.05) are depicted on graphs. Images are representative of four young intact male dog prostates. High resolution images shown here and others from biological replicates are available through the GUDMAP database at https://doi.org/10.25548/16-WMM4.

Collagen accumulation is often linked to inflammation. [40] We examined the frequency of hematolymphoid cells within each prostatic region and the continuous distribution across the proximodistal axis of young intact male prostate (S4 and S5 Figs). We found that hematolymphoid cells are not distributed in a regional or proximodistal pattern but rather distributed homogenously throughout the prostate.

We tested whether aging increases the number of hematolymphoid cells in the peripheral and capsule regions of old versus young intact males. We stained sections of prostate from old

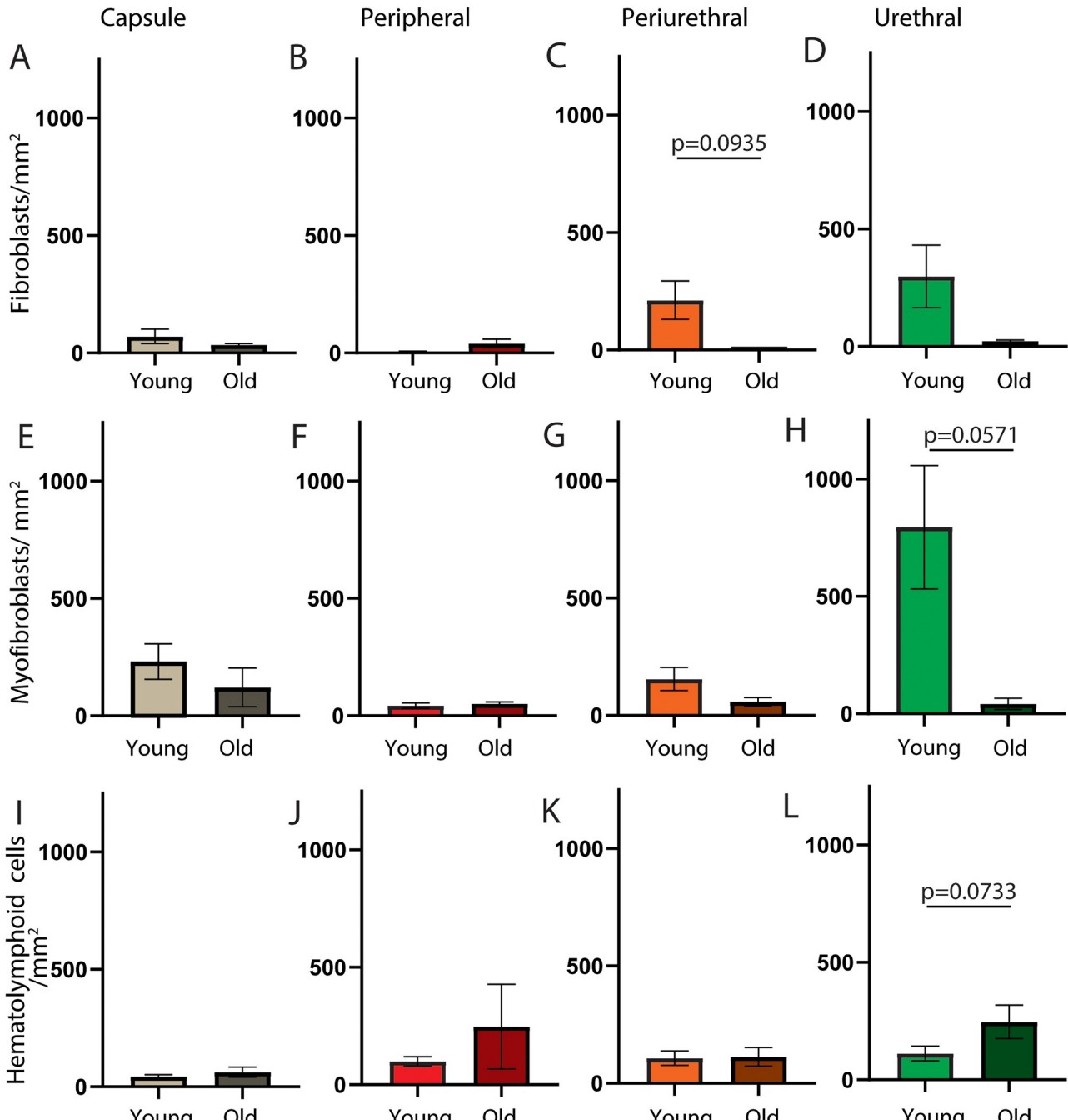

**Fig 4. Aging does not change myofibroblast, fibroblast, and hematolymphoid cell densities in the intact male dog prostate.** Multiplex immunostaining with antibodies against protein tyrosine phosphatase, receptor type C (PTPRC), smooth muscle actin (ACTA2), fibroblast specific protein (S100A4) and vimentin (VIM) was performed on complete transverse sections of young (≤3 years old) and old (≥5 years old) intact male canine prostates. Nuclei were stained with DAPI. Densities of (A-D) hematolymphoid cells (PTPRC+), (E-H) fibroblasts (S100A4+, ACTA2-, VIM+) and (I-L) myofibroblasts (S100A4+, ACTA2+, VIM+) were determined for the capsule, peripheral, periurethral, and urethral regions. Results are mean ± SE, five to eight young males and three to four old males per group. Student's t-test was used to compare cell densities between young and old intact males. Cell densities did not significantly differ (p<0.05) between young and old males in any of the prostatic regions examined. P-values between 0.1 and 0.05 are listed. High resolution images shown here and others from biological replicates are available through the GUDMAP database at https://doi.org/10.25548/16-WMM4.

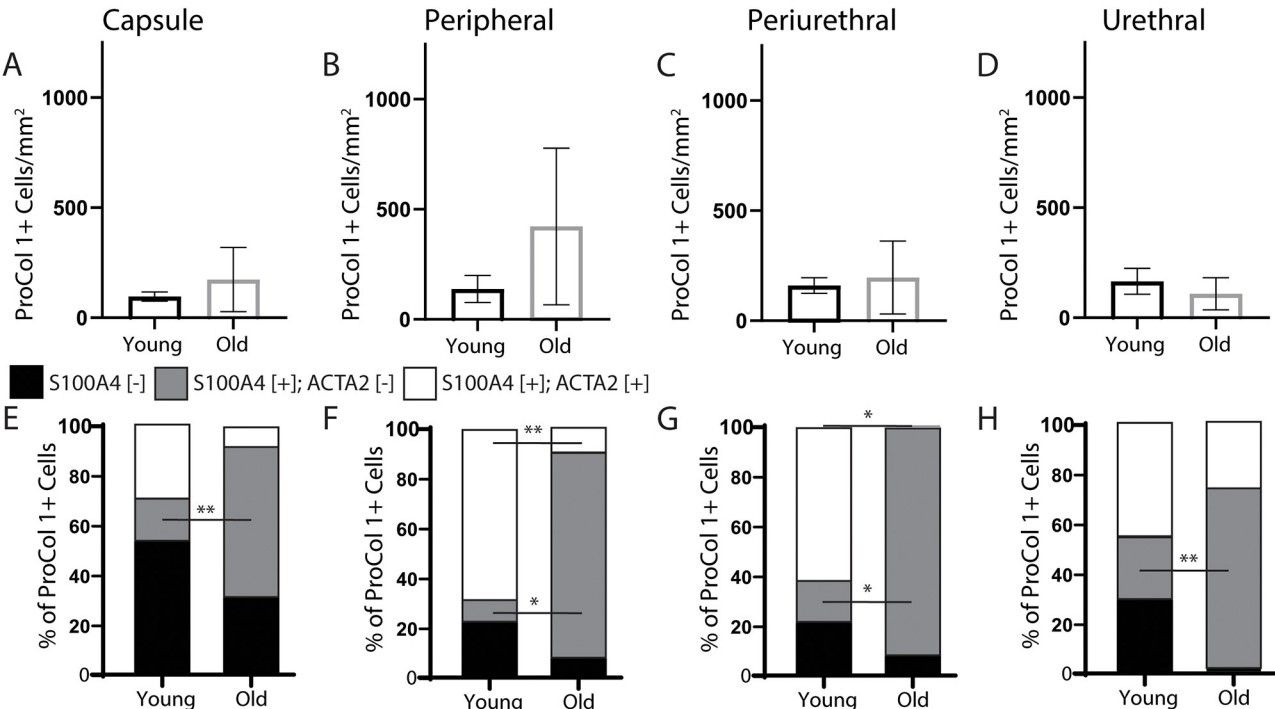

**Fig 5. Aging shifts prostatic procollagen 1A1 production in intact male dog prostate from myofibroblasts to fibroblasts.** Multiplex immunostaining with antibodies against procollagen 1A1 (ProCol 1A1), smooth muscle actin (ACTA2), and fibroblast specific protein (S100A4) was performed on complete transverse sections of young (≤3 years old) and old (≥5 years old) intact male canine prostate. Nuclei were stained with DAPI. (A-D) The overall density of ProCol 1A1+ cells was determined in the capsule, peripheral, periurethral, and urethral regions. (E-H) Cell identities of prostatic ProCol 1A1+ cell types were determined in the capsule, peripheral, periurethral, and urethral regions. Results are mean ± SE, five young males and four old males per group. Student's t-test was used to compare overall ProCol 1A1+ cell density and density of specific ProCol1A1+ cell types between young and old. A single asterisk identifies a p-value between 0.01 and 0.05 and a double asterisk identifies a p-value less than 0.01. High resolution images shown here and others from biological replicates are available through the GUDMAP database at https://doi.org/10.25548/16-WMM4.

intact male dogs and performed regional counts of hematolymphoid cells (PTPRC+). The number of hematolymphoid cells in the prostatic capsule and peripheral regions does not significantly differ between old and young intact males (Fig 4).

### 3.3. Aging shifts the predominant prostatic procollagen 1A1 producing cells from myofibroblasts to fibroblasts

We stained sections from young and old intact male dogs with procollagen 1A1 (ProCOL1A1), smooth muscle actin (ACTA2), and fibroblast specific protein (S100A4) to identify collagen synthesizing cells. The overall frequency of ProCOL1A1+ cells in the prostatic capsule and peripheral regions does not significantly differ between young and old intact males (Fig 5A–5D). There is, however, an aging-related shift in the identity of procollagen 1A1 producing cells. Procollagen 1A1 producing S100A4+, ACTA2+ cells (prostatic myofibroblasts) are more abundant in young dogs, while procollagen 1A1 producing S100A4+, ACTA2- (prostatic fibroblasts) are more abundant in old dogs (Fig 5E–5H).

## 4. Discussion

We adapted an immunohistochemical prostate cell identification key, previously published for use in mice, [21] for use in dogs. We successfully stained dog prostate sections with the same

antibodies and same antibody combinations previously used to identify mouse prostatic cell types, enabling future comparisons between these two model species. We also determined the abundance and distribution of individual cell types to enable comparisons to human prostate [13, 14] and address whether canine prostate has a homogeneous [5, 39] or heterogenous [38] composition. Our activities establish a baseline cellular map of the canine prostate that will be used to determine the impact of age, inflammation and other disease processes on prostatic cellular composition and to ultimately identify disease causing cell types. Our results indicate that canine prostate cell composition is heterogeneous, consistent with older descriptions of the canine prostate in which epithelial cells predominate in the peripheral region and stromal cells predominate in the periurethral region. [38] The peripheral and periurethral regions are not delineated by a visible anatomic or histologic boundary. We defined them by dividing the glandular region into two concentric rings of equal thickness. Our approach led to the discovery that the peripheral and periurethral regions differ in epithelial and stromal cell composition (Figs 1 and 3) and propensity to develop aging-related collagen accumulation [12]. Our results support the notion that peripheral and periurethral region microenvironments uniquely regulate cell behavior and cell composition and may even confer differences in aging-related pathology, much as the human prostate transition and peripheral zones differ in susceptibility to benign hyperplasia and cancer. [41]

We show here that canine prostate epithelial cells are not equally distributed between prostatic zones and specifically that basal cells are more frequent in the canine prostatic urethra. Epithelial cell distribution has not been quantified in the mouse prostate, but published images and anecdotal observations hint towards increased basal cells in the mouse prostatic urethra. [21] Henry et. Al. recently showed enriched epithelial cells in the peripheral zone of human prostate compared to transition/central zones in fluorescence-activated cell sorting. They also include whole transverse sections of human prostate stained for KRT5 and KRT8 with more visually prominent KRT5 staining in the urethra. [13]

Activated myofibroblasts mediate fibrosis in many organ systems. [15–20] Even though we noted previously that aging causes collagens to accumulate in the capsule and peripheral regions of intact canine prostate, [12] we found no evidence that aging causes myofibroblasts to become more frequent or initiate pro-collagen 1A1 synthesis in these prostatic regions. Instead, aging shifted the most dominant procollagen 1A1 positive stromal cell type from prostatic myofibroblasts to fibroblasts. Because prostatic collagens accumulate with age, and because we did not observe a net increase in overall quantity of prostatic collagen producing cells, we consider the aging-related transition in the identity of collagen producing cells to be of probable significance with respect to prostatic collagen homeostasis. It is possible that the procollagen 1A1 producing fibroblasts in the older male prostate are more efficient at synthesizing collagens than procollagen 1A1 producing myofibroblasts in young dogs and therefore drive pathological collagen accumulation. We will explore this hypothesis in future studies.

The identity of collagen producing cells has clinical significance with respect to pharmacological targeting of collagen biosynthesis pathways within the aging prostate. We therefore considered two possible mechanisms by which aging could drive a change in collagen producing cell identities. First, it is possible the procollagen 1A1 producing fibroblasts represent a unique but rare cell lineage in young intact male prostate and is driven by the aging process to proliferate and become more numerous. A second possibility is that procollagen 1A1 positive prostatic fibroblasts in older intact males derive from smooth muscle actin positive myofibroblasts and undergo an aging dependent loss of smooth muscle actin expression. Our results favor the second possibility, as we did not observe a corresponding loss of myofibroblasts with age. Prospective studies are needed to determine order of events and causal relationships.

It is important to note that we used a small but diverse sampling of dog prostates to collect baseline information of the canine prostate that is generalizable to the intact male canine population as a whole. We did not obtain sufficient numbers to stratify our population by factors such as breed, body mass, or prostate volume, however this is an important future direction. Future studies are needed to understand the intersection of inflammation, collagen accumulation, and other factors contributing to prostatism including body mass, prostate volume, and breed in dogs and race in humans.

## 5. Conclusions

Our results concur with older studies of dog prostate documenting a heterogeneous rather than homogenous histologic structure across the capsule, peripheral, periurethral and urethral regions. Our results also suggest that prostatic myofibroblasts, unlike myofibroblast in other tissues, do not expand with age and do not increase collagen production with age in intact male dogs. Finally, we identified aging-related shift in cell types responsible for collagen production that could play an important role in collagen accumulation and urinary dysfunction in aging males. All images from this project are available in the GUDMAP database at https://doi.org/10.25548/16-WMM4.

## Supporting information

**S1 Fig. Prostatic regions in canine prostate.** Shown is a tile scan of a hematoxylin and eosin stained prostate of a young intact male dog overlaid with regions (left) and shown in the same scale (right). The urethral region is defined by the urethral epithelium and its associated stroma (circumferential orientation of stromal fibers). The capsule is the stroma including large vessels and large bundles of muscle comprising the outer boundary of the prostate and is void of epithelial cells. The periurethral and peripheral regions do not have clear anatomical boundaries and are defined by dividing the tissue between the capsule and urethra into two rings. This particular prostate is a rare example of a (*) thin band of stroma running circumferentially and separating the two zones in the dorsal portion of the gland). High resolution images shown here and others from biological replicates are available through the GUDMAP database at https://doi.org/10.25548/16-WMM4.
(TIF)

**S2 Fig. Perivascular cells in young intact canine prostate.** Multiplex immunostaining with antibodies against smooth muscle actin (ACTA2), Platelet derived growth factor receptor beta (PDGFRB) and Platelet and endothelial cell adhesion molecule (PECAM) was performed on complete transverse sections of the young ($\leq$3 years old) intact male canines. Nuclei were stained with DAPI. (A-C) Endothelial cells, pericytes, and perivascular smooth muscle cells were identified based on combinatorial staining. Endothelial cells, pericytes, and perivascular smooth muscle cells were visualized within each region (capsule depicted in D). Images are representative of 9 young intact male dog prostates. High resolution images shown here and others from biological replicates are available through the GUDMAP database at https://doi.org/10.25548/16-WMM4.
(TIF)

**S3 Fig. Tissue architecture in young vs old dogs.** Sections from young and old intact male dogs were stained with hematoxylin and eosin. Results are representative of 9 young and 4 old intact male dog prostates. High resolution images shown here and others from biological replicates are available through the GUDMAP database at https://doi.org/10.25548/16-WMM4.
(TIF)

**S4 Fig. Hematolymphoid cell density is not altered by region.** Sections from young intact male dogs were immunostained with an antibody against protein tyrosine phosphatase, receptor type C (PTPRC). Nuclei were stained with DAPI. Hematolymphoid cells (PTPRC+) were visualized, and densities quantified within each region. Results are mean ± SE of 6–8 young intact male dog prostates, a one-way ANOVA was used to compare regions, no statistical differences were found between regions. High resolution images shown here and others from biological replicates are available through the GUDMAP database at https://doi.org/10.25548/16-WMM4.
(TIF)

**S5 Fig. Proximal to distal distribution of hematolymphoid cells.** Four prostates from young intact males were tile scanned in three 20X field tall strip from the urethra to the capsule in the right or left lateral portion of the gland (A-D). We used image J- plot profile to determine the average gray value for each column of pixels, creating a plot of the urethral to capsule distribution of hematolymphoid cells in the prostate (A-D). Using this technique, we found that hematolymphoid cells were not distributed in a proximal to distal pattern but rather distributed fairly ubiquitously throughout the prostate. High resolution images shown here and others from biological replicates are available through the GUDMAP database at https://doi.org/10.25548/16-WMM4.
(TIF)

## Author Contributions

**Conceptualization:** Hannah Ruetten, Kyle A. Wegner, Chad M. Vezina.

**Data curation:** Hannah Ruetten, Clara Cole, Marlyse Wehber, Nelson T. Peterson, Brandon R. Scharpf, Chad M. Vezina.

**Formal analysis:** Hannah Ruetten, Clara Cole, Marlyse Wehber, Nicholas M. Girardi, Nelson T. Peterson.

**Funding acquisition:** Chad M. Vezina.

**Investigation:** Hannah Ruetten, Clara Cole, Marlyse Wehber.

**Methodology:** Hannah Ruetten, Kyle A. Wegner, Chad M. Vezina.

**Project administration:** Hannah Ruetten, Chad M. Vezina.

**Resources:** Hannah Ruetten, Kyle A. Wegner, Michael F. Romero, Michael W. Wood, Sara A. Colopy, Chad M. Vezina.

**Supervision:** Hannah Ruetten, Dale E. Bjorling, Chad M. Vezina.

**Validation:** Hannah Ruetten, Clara Cole, Marlyse Wehber, Chad M. Vezina.

**Visualization:** Hannah Ruetten, Clara Cole, Marlyse Wehber, Chad M. Vezina.

**Writing – original draft:** Hannah Ruetten, Clara Cole, Marlyse Wehber, Chad M. Vezina.

**Writing – review & editing:** Hannah Ruetten, Clara Cole, Marlyse Wehber, Kyle A. Wegner, Nicholas M. Girardi, Nelson T. Peterson, Brandon R. Scharpf, Michael F. Romero, Michael W. Wood, Sara A. Colopy, Dale E. Bjorling, Chad M. Vezina.

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
