## [Decision Letter · Decision Letter 0]

4 Jun 2020

PONE-D-20-11028

An immunohistochemical prostate cell identification key indicates that aging shifts procollagen 1A1 production from myofibroblasts to fibroblasts in dogs prone to prostate-related urinary dysfunction

PLOS ONE

Dear Dr. Vazina:

Thank you for submitting your manuscript to PLOS ONE. After careful consideration, we feel that it has merit but does not fully meet PLOS ONE’s publication criteria as it currently stands. Therefore, we invite you to submit a revised version of the manuscript that addresses the points raised during the review process.

**Reviewer #1: **

The authors present interesting data that show a shift in the collagen-producing cells of the prostate in dogs, from a myofibroblastic cell type to a mature fibroblastic cell type. The results are important because, if confirmed, this information can be used to understand how fibrosis in aging prostates in a dog may be useful in our understanding of this important pathological phenomenon in humans. The findings are also important because previous assumptions within the field were that there was a cell type shift behind changes in collagen deposition in the prostate; the findings of this paper suggest that it is not a cell type shift in numbers, but rather a shift in which cells produce the collagen. Therefore the findings of this paper may represent a shift in the paradigm on how we think of fibrosis and collagen deposition in the stromal compartment of the prostate. We look forward to future papers in this area building upon this information.

Minor comments for the authors to consider in the revision of this manuscript:

1. the IHC figures should include higher power images in addition to the broader versions shown in the figures. It is appreciated that the lower power views allow a broad view, but additions of higher power images will greatly assist the readers' abilities to see how the cell types are arranged with respect to each other. They will give a good view of architectural changes. In addition, H&E images should be included for this same purpose, and to reveal any possible developing pathologies from these collagen changes.

2. For the data presented in Figures 3 (panels H and I), figure 4, and Figure 5 (Panels A through D), would it be a better representation to show these data as a percent of total cells rather than per image? There is a very large variability in some of these data, and perhaps an adjustment to the number of total cells might allow for reducing this across groups, and perhaps finding differences where p values currently outside the normally-accepted level of "difference" could be reduced to difference. IN addition, this would actually be a more valid way to assess the figures anyway.

3. The authors could comment on the body mass of these dogs: is there any difference in obesity rates or other obvious BPH-related risk factors in some of the older animals that could play a role in collagen deposition and cell type shifts as the animals age? The group has a great resource of animals that could yield this information. If the numbers are too low currently to warrant further stratification, perhaps a comment on this would be warranted.

**Reviewer #2: **

The work by Ruetten *et al *looks at the identification of stromal cells responsible for the production of procollagen 1A1 in dogs with lower urinary tract dysfunction related to prostate changes associated with aging. The authors used immunohistochemical analysis to identify the location of these cells. This study provides a framework for future studies when looking at changes in prostatic diseases in dogs. However, there are a few questions that require clarification.

• Our current understanding of the human prostate anatomy is based on the classic work by McNeal (1968) who proposed four different prostatic zones based on their anatomic location and histological features. The presence of these four zones have been validated recently by others (Selman et al 2011, Samson 2012) and also using MRI technologies (Yacoub 2018). Please revised the statement “…that are organized into three zones” (lane 71) accordingly.

• Please revise the following statement for accuracy: “We found that aging increases collagen fiber thickness and density in the capsule and peripheral regions intact dogs susceptible to prostatism”. The conclusions of this reference suggest that older dogs have increased collagen fiber thickness and density compared to young male dogs. However, in this particular paper there was not any evidence that these changes in collagen were present in dogs with “prostatism”. Dogs were euthanized for reasons other than prostatic disease. Unlike humans, typically no clinical signs of LUTS are present in dogs unless the BPH condition progress to hematuria, tenesmus, urethral discharge or stilted gait. Similarly, in prostate cancer signs and symptoms are also present later in the disease.

• In figure 1 the data on epithelial cells is presented as a percentage over total number of cells, however from figure 2, number of cells/image. Given that each field is representative of the gland architecture, perhaps it will be more informative if the data is presented as percentage of total number of cells, also for consistency.

• Did the authors compared whether cell densities for each of these populations are different between the four regions? Looking at Figure 4 , it looks like there are more fibroblasts in the capsule and peripheral zone compared to the periurethral and urethral zones. Unfortunately, it is hard to determine because the y axis used are different between groups.

• Figure 5 (panels E to H) will benefit with the use of colors instead of the shades of gray. Another alternative is to use a combination of white, gray and black.

• The authors did a great job looking at epithelial and fibroblasts cell populations using a combination of markers to classify them (luminal/intermediate/basal and NE for epithelial cells and fibroblasts/myofibroblasts) and only use one marker for the hematolymphoid (?), perhaps one of the most well studied and diverse in terms of subpopulations. This study will be complete if the authors can provide some evidence of lymphoid and/or myeloid cell populations distribution across the dog prostate. Given that inflammatory processes are clearly involved in the promotion of collagen deposition (per the authors premise for looking at collagen in this study), determining whether macrophages or lymphocytes associate with areas of high collagen content or presence of specific fibroblasts population will give a much better picture on hematolymphoid cells in young vs old dogs.

• It is well known that larger breeds of dogs are at higher risk of developing prostate cancer. Did the authors compared whether changes in collagen deposition are different between different breeds?. Or if prostate size was associated with the composition/distribution of collagen?

We look forward to receiving your revised manuscript.

Kind regards,

Mohammad Saleem., Academic Editor

PLOS ONE

Masonic Cancer Center, University of Minnesota

---

## [Author Response · Author response to Decision Letter 0]

6 Jul 2020

Response to reviewer comments for “An immunohistochemical prostate cell identification key indicates that aging shifts procollagen 1A1 production from myofibroblasts to fibroblasts in dogs prone to prostate-related urinary dysfunction”.

Reviewer #1: 

The authors present interesting data that show a shift in the collagen-producing cells of the prostate in dogs, from a myofibroblastic cell type to a mature fibroblastic cell type. The results are important because, if confirmed, this information can be used to understand how fibrosis in aging prostates in a dog may be useful in our understanding of this important pathological phenomenon in humans. The findings are also important because previous assumptions within the field were that there was a cell type shift behind changes in collagen deposition in the prostate; the findings of this paper suggest that it is not a cell type shift in numbers, but rather a shift in which cells produce the collagen. Therefore, the findings of this paper may represent a shift in the paradigm on how we think of fibrosis and collagen deposition in the stromal compartment of the prostate. We look forward to future papers in this area building upon this information.

Comment 1: The IHC figures should include higher power images in addition to the broader versions shown in the figures. It is appreciated that the lower power views allow a broad view, but additions of higher power images will greatly assist the readers' abilities to see how the cell types are arranged with respect to each other. They will give a good view of architectural changes. In addition, H&E images should be included for this same purpose, and to reveal any possible developing pathologies from these collagen changes.

Response: High power inlays have been added to Figures 1, 2, and 3. High resolution images contributing to the figures have been made freely accessible through the GUDMAP database. A supplemental figure has been added containing representative H&E images of old vs young intact dogs in all regions. Additional H&E images of all regions for all dog prostates used in this and our previous study have been made freely accessible through the GUDMAP database. A clickable digital object identifier is included in the text to direct readers to the GUDMAP database, where a library of images corresponding to this manuscript are stored. Note that the GUDMAP database contains all of the original data used for the figures – not just the representative images that are shown, but all other data used to draw the conclusions. We include all raw data to enhance rigor and transparency.

Comment 2: For the data presented in Figures 3 (panels H and I), figure 4, and Figure 5 (Panels A through D), would it be a better representation to show these data as a percent of total cells rather than per image? There is a very large variability in some of these data, and perhaps an adjustment to the number of total cells might allow for reducing this across groups, and perhaps finding differences where p values currently outside the normally-accepted level of "difference" could be reduced to difference. IN addition, this would actually be a more valid way to assess the figures anyway.

Response: To be consistent with our prior study we have used cells per area as our unit of measurement for all stromal cell types. However, we agree that “per image” isn’t as precise and have converted figures quantifying stromal cells to cells per mm2.

Comment 3: The authors could comment on the body mass of these dogs: is there any difference in obesity rates or other obvious BPH-related risk factors in some of the older animals that could play a role in collagen deposition and cell type shifts as the animals age? The group has a great resource of animals that could yield this information. If the numbers are too low currently to warrant further stratification, perhaps a comment on this would be warranted.

Response: We did not have sufficient numbers for stratification but have added this to the discussion as a future direction.

Reviewer #2: 

The work by Ruetten et al looks at the identification of stromal cells responsible for the production of procollagen 1A1 in dogs with lower urinary tract dysfunction related to prostate changes associated with aging. The authors used immunohistochemical analysis to identify the location of these cells. This study provides a framework for future studies when looking at changes in prostatic diseases in dogs. However, there are a few questions that require clarification.

Comment 1: Our current understanding of the human prostate anatomy is based on the classic work by McNeal (1968) who proposed four different prostatic zones based on their anatomic location and histological features. The presence of these four zones have been validated recently by others (Selman et al 2011, Samson 2012) and also using MRI technologies (Yacoub 2018). Please revised the statement “…that are organized into three zones” (lane 71) accordingly.

Response: This has been corrected to “four zones”.

Comment 2: Please revise the following statement for accuracy: “We found that aging increases collagen fiber thickness and density in the capsule and peripheral regions intact dogs susceptible to prostatism”. The conclusions of this reference suggest that older dogs have increased collagen fiber thickness and density compared to young male dogs. However, in this particular paper there was not any evidence that these changes in collagen were present in dogs with “prostatism”. Dogs were euthanized for reasons other than prostatic disease. Unlike humans, typically no clinical signs of LUTS are present in dogs unless the BPH condition progress to hematuria, tenesmus, urethral discharge or stilted gait. Similarly, in prostate cancer signs and symptoms are also present later in the disease.

Response: This has been corrected for accuracy.

Comment 3: In figure 1 the data on epithelial cells is presented as a percentage over total number of cells, however from figure 2, number of cells/image. Given that each field is representative of the gland architecture, perhaps it will be more informative if the data is presented as percentage of total number of cells, also for consistency.

Response: To be consistent with our prior study we have used cells per area as our unit of measurement for all stromal cell types. However, we agree that “per image” isn’t as precise and have converted figures quantifying stromal cells to cells per mm2.

Comment 4: Did the authors compared whether cell densities for each of these populations are different between the four regions? Looking at Figure 4 , it looks like there are more fibroblasts in the capsule and peripheral zone compared to the periurethral and urethral zones. Unfortunately, it is hard to determine because the y axis used are different between groups.

Response: Statistical comparisons were only made between or among images collected form the same microscope using the same imaging settings and magnification. And were only made for cell types in a single stain. However, we have made y-axis and units consistent across figures to enable visual comparisons between groups.

Comment 5: Figure 5 (panels E to H) will benefit with the use of colors instead of the shades of gray. Another alternative is to use a combination of white, gray and black.

Response: This figure now utilizes white, gray, and black.

Comment 6: The authors did a great job looking at epithelial and fibroblasts cell populations using a combination of markers to classify them (luminal/intermediate/basal and NE for epithelial cells and fibroblasts/myofibroblasts) and only use one marker for the hematolymphoid (?), perhaps one of the most well studied and diverse in terms of subpopulations. This study will be complete if the authors can provide some evidence of lymphoid and/or myeloid cell populations distribution across the dog prostate. Given that inflammatory processes are clearly involved in the promotion of collagen deposition (per the authors premise for looking at collagen in this study), determining whether macrophages or lymphocytes associate with areas of high collagen content or presence of specific fibroblasts population will give a much better picture on hematolymphoid cells in young vs old dogs.

Response: We chose stain combinations based on our immunohistochemical identification key for prostatic cell types in mouse in order to allow cross species comparisons. We agree that many subpopulations exist within our categories, not only in the hematolymphoid category but in other cell type classifications as well. We hope that this work is a starting point for researchers who can add on their own immunohistochemical stain combinations and classify further unique subpopulations as part of a partonomic ontology. We agree that studies assessing the inflammatory environment in which collagen changes are occurring in an important future direction but rather than adding more to this body of work we think this information would be better suited for a standalone manuscript where there is room to explore not only lymphoid vs myeloid but common immune cell types including B-cells, T-cells, neutrophils, macrophages, etc. and their known subtypes.

Comment 7: It is well known that larger breeds of dogs are at higher risk of developing prostate cancer. Did the authors compared whether changes in collagen deposition are different between different breeds?. Or if prostate size was associated with the composition/distribution of collagen?

Response: We did not have sufficient numbers for stratification but have added this to the discussion as a future direction

---

## [Editor Report · Decision Letter 1]

10 Jul 2020

An immunohistochemical prostate cell identification key indicates that aging shifts procollagen 1A1 production from myofibroblasts to fibroblasts in dogs prone to prostate-related urinary dysfunction

PONE-D-20-11028R1

Dear Dr. Chad M Vezina

We’re pleased to inform you that your manuscript has been judged scientifically suitable for publication and will be formally accepted for publication once it meets all outstanding technical requirements.

Kind regards,

**Mohammad Saleem,  **

**Dept. of Urology, Masonic Cancer Center, University of Minnesota**

---

## [Editor Report · Acceptance letter]

15 Jul 2020

PONE-D-20-11028R1 

An immunohistochemical prostate cell identification key indicates that aging shifts procollagen 1A1 production from myofibroblasts to fibroblasts in dogs prone to prostate-related urinary dysfunction 

Dear Dr. Vezina:

I'm pleased to inform you that your manuscript has been deemed suitable for publication in PLOS ONE. Congratulations! Your manuscript is now with our production department. 

Kind regards, 

on behalf of

Dr. MOHAMMAD Saleem 

Academic Editor

PLOS ONE